# Free-Running Single-Photon Detection via GHz Gated InGaAs/InP APD for High Time Resolution and Count Rate up to 500 Mcount/s

**DOI:** 10.3390/mi14020437

**Published:** 2023-02-12

**Authors:** Wen Wu, Xiao Shan, Yaoqiang Long, Jing Ma, Kun Huang, Ming Yan, Yan Liang, Heping Zeng

**Affiliations:** 1School of Optical-Electrical and Computer Engineering, University of Shanghai for Science and Technology, Shanghai 200093, China; 2State Key Laboratory of Precision Spectroscopy, East China Normal University, Shanghai 200062, China; 3Chongqing Institute of East China Normal University, Chongqing 401147, China; 4Shanghai Research Center for Quantum Sciences, Shanghai 201215, China

**Keywords:** single-photon detector, avalanche photodiode

## Abstract

Free-running InGaAs/InP single-photon avalanche photodiodes (SPADs) typically operate in the active-quenching mode, facing the problems of long dead time and large timing jitter. In this paper, we demonstrate a 1-GHz gated InGaAs/InP SPAD with the sinusoidal gating signals asynchronous to the incident pulsed laser, enabling free-running single-photon detection. The photon-induced avalanche signals are quenched within 1 ns, efficiently reducing the SPAD’s dead time and achieving a count rate of up to 500 Mcount/s. However, the timing jitter is measured to be ~168 ps, much larger than that of the SPAD with synchronous gates. We adjust the delay between the gating signals and the incident pulsed laser to simulate the random arrival of the photons, and record the timing jitter, respectively, to figure out the cause of the jitter deterioration. In addition, the effects of the incident laser power and working temperature of the APD on the time resolution have been investigated, broadening the applications of the GHz gated free-running SPAD in laser ranging and imaging, fluorescence spectroscopy detection and optical time-domain reflectometry.

## 1. Introduction

Recent years have seen single-photon detectors (SPDs) develope dramatically with the boom in quantum information technology [1,2]. Superconducting nanowire single-photon detectors (SNSPDs) are already close to 100% detection efficiency with dark counts less than 10 cps [3,4]. However, the need of cryogenic operating temperatures limits the SNSPD’s applications. InGaAs/InP single-photon avalanche photodiodes (SPADs) provide a valid alternative by working at room temperature or slightly below, proving to be an excellent candidate for single-photon detection in the near infrared. InGaAs/InP SPADs possess the features of easy integration and compact construction, and are increasingly employed in applications such as laser imaging detection and ranging, deep space optical communication, and fluorescence spectroscopy detection [5,6,7,8].

Typically, InGaAs/InP avalanche photodiodes (APDs) are operated in the active-quenching mode to achieve free-running photon detection [9,10,11]. The APD is immediately quenched after the avalanche is detected by actively reducing the APD’s bias voltage. It usually takes tens of nanoseconds for the SPAD to recover for the next detection, which we call dead time, resulting in a maximum count rate of the order of MHz. For applications such as deep space optical communication, the relatively low count rate directly limits the communication speed. The use of SPAD arrays could solve this problem; however, either adding to the system’s complexity or increasing the detector’s error counts [12]. In this paper, we propose a GHz-gated InGaAs/InP SPAD for free-running single-photon detection. The GHz gating signals combine with the DC voltage to bias the APD, switching the APD between the over and below of the breakdown voltage. In this mode, the SPAD could resume detecting the next incident photon in less than 1 ns, reducing the dead time and increasing the maximum count rate [13,14]. Moreover, the error counts, especially the afterpulses, could be effectively reduced by shortening the avalanche duration. Different from the traditional InGaAs/InP SPD in the gated Geiger mode, the gating signal of the free-running APD is not synchronized with the incident laser pulse. The detection efficiency undoubtedly decreases since some laser signals don’t arrive within the gating signals and are not detected. Increasing, the duty cycle of the gates is a valid option [15]. Timing jitter is another issue that cannot be ignored for free-running InGaAs/InP APD. In general, the timing jitter exceeds 100 ps, which is much larger than that in the synchronous gated mode [16,17,18].

In this paper, we demonstrate a 1-GHz sinusoidally gated free-running InGaAs/InP APD. To eliminate the spike noise produced by charging and discharging of the GHz gating signals on the APD, numerous techniques, including self-differencing, sinusoidal gating, and the combination of the two, have been put forth so far [19,20,21]. The spike noise’s suppression ratio could be greater than 30 dB. Here, we choose the low-pass filtering technique by employing a low-pass filter with a cutoff frequency of 700 MHz. The avalanche signal could be effectively extracted with the spike noise suppressed down to the thermal noise level. The dark count rate is only 1.5 kHz with the afterpulse probability of ~11.5% at the detection efficiency of 5.0% while the InGaAs/InP APD is Peltier cooled to −30 ℃. The maximal count rate of the SPD, which is measured to be ~500 MHz when illuminated by a pulsed laser source with the repetition frequency of 500 MHz, is significantly higher than that of earlier free-running SPDs [22,23]. The SPD’s timing jitter is recorded to be 76 ps in the synchronized gated mode and 168 ps in the free-running mode. In order to investigate the reasons for the increase in the timing jitter, we adjust the delay between the 1-GHz sinusoidal gate and the incident optical signal, compute the avalanche count and time distribution to simulate the random arrival of the incident photons. In addition to the intensity of the incident light, APD’s operation temperature is changed to evaluate the effects on the time resolution, providing effective detection support for laser ranging, optical time domain reflectometry, and other high time-resolved application needs.

## 2. Experimental Setup

The schematic setup of the 1-GHz sinusoidally gated free-running InGaAs/InP SPAD is depicted in Figure 1. The sinusoidal signal is generated by the synthesized signal generator (SSG-6000 RC, Mini-circuits) and then amplified by an RF amplifier with a fixed gain of 25 dB (ZHL-5W- 422+, Mini-Circuits). Considering the frequency distributions of the avalanche and spike noise, a low-pass filter (LPF) with attenuation higher than 40 dB at 1 GHz and a cutoff frequency of 700 MHz is used to obtain the avalanche signal. To ensure the signal-to-noise ratio (SNR), we employ a high-pass filter to remove the sideband noise of the amplified 1-GHz sinusoidal gates. The output of the LPF is coupled to a low-noise amplifier, and a digital oscilloscope (DSA70404C, Tektronix) is utilized to capture the amplified avalanche signal. The InGaAs/InP APD is fiber-pigtailed with the effective detecting diameter of 45 μm, and the DC bias voltage could be changed to obtain different detection efficiencies. A four-stage thermal electric cooler (TEC) is used to cool the APD. By altering the current flowing into the TEC, the operating temperature of the APD could be varied from room temperature to −60 °C. To assess the performance of the SPAD, we employ a home-made 1550-nm pulsed laser with a pulse width of about 50 ps as the incident light source. The repetition frequency of the laser pulses could increase from kHz to GHz with the external trigger signal. Furthermore, the rise time of the laser pulse is measured to be ~20 ps with the timing jitter of ~2 ps.

First, we set the SPAD’s operating temperature to −30 °C. The incident laser is attenuated by a digital attenuator to contain 0.1 photons/pulse on average at 10 MHz. Figure 2 shows the photon detection efficiency (PDE), dark count rate (DCR) and probability of the afterpulse (P-afterpulse) as a function of the dc bias voltage. Here, the amplitude of the sinusoidal is fixed at 10 V. The three curves exhibit a rising tendency as the dc voltage increases. The PDE increases approximately linearly, while the DCR gets higher slowly and remains at the level of kHz. As for the P-afterpulse, it goes up slightly when the PDE is lower than 5.0%, and then increases sharply. The P-afterpulse rises up to 45.0% at the PDE of 8.6%, while the DCR is 1.8 kHz, limiting the SPAD’s applications in quantum information [24,25,26,27]. Introducing an additional dead time, i.e., turning the gating signal or counting output off for a period of time after the avalanche signal is detected, can validly solve this problem. Higher PDE and lower error counts, including dark counts and afterpulses, could be predicted.

## 3. SPAD’s Maximum Count Rate

In the experiment, we use the LPF, which cuts off at 700 MHz, to extract the photon-induced avalanche signal, inevitably filtering out the avalanche signal’s high-frequency components. As a result, the output of the SPAD has a longer rise time, which lowers its maximum count rate. Hence, we tune the repetition frequency of the laser (f_laser_) to 500 MHz to illuminate the SPAD to determine the maximum count rate. Figure 3a exhibits the count rates versus incident photon flux at three different PDEs. First, they grow linearly with almost the same slope. While the flux exceeds 1 photon/pulse, the rising trends slow down, whereas the differences of the count rates at different PDEs decrease, which may be caused by the increase of the multi-photon’s probability. Finally, all the count rates reach the maximum of approximately 500 MHz. The SPAD’s output waveform is recorded by the oscilloscope, as shown in the inset of Figure 3a. The illumination flux is set to be 100 photons/pulse, while the residual background is recorded under no illumination. The amplitude of the avalanche signal is ~530 mV, while that of the background noise is ~18 mV, verifying the high SNR of the detecting scheme. The duration of the avalanche is ~2 ns, corresponding to the maximum count rate of 500 MHz. Employing sinusoidal gates with higher repetition frequencies (f_gate_) and low-pass filters with higher cutoff frequencies may be an effective means of increasing the SPAD’s maximum count rate. Figure 3b plots the simulated count rates as a function of the incident photon flux by a theoretical calculation, using the PDE of 5.0% and DCR of 1.8 kHz. The measured curve agrees well with the simulated one when the incident flux is lower than 1 photon/pulse, both showing a linear variation. With stronger illumination, the two curves start to deviate, probably caused by the increase of the afterpulses. Finally, the two curves saturate at 500 MHz, verifying the maximum count rate of the free-running SPAD.

## 4. SPAD’s Timing Jitter

The timing jitter is another key parameter for single-photon detection, and is especially important for applications such as laser ranging and imaging. The TCSPC (PicoHarp 300, PicoQuant) with the minimum resolution of 4 ps is used in the experiment. A synchronizing signal of the laser’s trigger is sent to the “START” of the TCSPC, while the output of the SPAD is sent to the “STOP”, resulting in the record of the SPAD’s timing jitter. Limited by the TCSPC’s maximum count rate, the f_laser_ is fixed at 10 MHz, while the intensity of the laser is set to contain 0.1 photon/pulse. We compare the jitter in the synchronously gating mode and in the free-running mode. First, the laser and the 1-GHz gates are synchronously triggered, and the delay between them are tuned to achieve the maximum count. The inset of Figure 4 displays the measured timing jitter with the full width at half maximum (FWHM) of 76 ps at the PDE of ~10.0%. Then the two trigger signals are switched to be unlocked to perform free-running single-photon detection, and the timing jitter is broadened to 168 ps. Moreover, we can see that the peak value of the count for the free-running SPAD is much lower than that for synchronously gating SPAD, indicating a decrease of the detection efficiency.

To analyze the cause of the degradation of the free-running SPAD’s time resolution, we lock the trigger signals of the sinusoidal gates and laser pulses, vary the delay between them at a certain interval, and record the corresponding time distribution of the SPAD’s output. By this method, the arrival of photons at arbitrary times can be well simulated. In the experiment, the signal sent to the “START” of the TCSPC are synchronized with the two triggers. Therefore, we change the delay between the synchronous signal (S_SYN_) and the trigger of the gates (S_gate_), and that between S_SYN_ and the trigger of the laser (S_laser_), respectively. Figure 5a displays the time distributions of the SPAD’s counts as a function of the delay between S_SYN_ and S_gate_. Considering the duration of the sinusoidal gates is 1 ns, the delay interval is set to 20 ps. At the maximum count rate, the delay is defined to be 0 ps, while the peak of the gate signal and that of the laser pulse are aligned. We steadily change the delay from −500 ps to 500 ps by the fixed 20 ps interval and record the SPAD’s timing jitters. From −500 ps to −200 ps, the count is approximately equal to the dark count, while the counting peak is not obvious, indicating that the photons cannot be detected. In this case, the photons are arriving at the bias voltage lower than the breakdown voltage. From −180 ps to 300 ps, the SPAD’s count gradually increases to the maximum, and then gradually drop to the dark-count level. When the delay is larger than 320 ps, the counting rate is kept approximately equal to dark count. We can conclude that the SPAD can respond to a single photon in ~500 ps, which is equal to half the period of a sinusoidal gating signal. The FWHM of each count distribution is approximately 76 ps, which is consistent with the result in the inset of Figure 4. Moreover, the appearance time of the counting peaks varies with the gate delay. We accumulate the time distributions with different S_gate_ delay and obtain an FWHM of ~180 ps, as shown in Figure 5b. In consideration of the 4-ps resolution of the TCSPC, the result is approximately equal to the timing jitter of the free-running SPAD, verifying that the random arrival of the gating signal is the cause of the larger timing jitter. We may decrease the width of the gating signals to improve the SPAD’s time resolution. Furthermore, the counting peaks are not symmetrically distributed around zero delay, which are more densely distributed in the part of the delay greater than 0 ps. As we all know, when the delay is greater than 0 ps, the peak of the sinusoidal gate arrives before the laser pulse. In order to understand the effect of the pre- and post-arrival times of the two signal peaks on the time resolution in more detail, we change the delay between the S_SYN_ and the S_laser_, and record the time distributions of the SPAD’s counts.

Figure 6a exhibits the time distributions of the SPAD’s counts versus the delay between S_SYN_ and S_laser_. We also set the delay to 0 ps when the maximum SPAD’s count is obtained. Similar to Figure 5a, from −500 ps to 500 ps, the count is first equal to the dark count, and then increases with the delay, reaches the maximum, and then declines gradually to the dark count. The counting of the peaks take place as the delay changes. However, the magnitude of the counting peaks’ shift seems much smaller than that in Figure 5a. Figure 6b shows the accumulation time distributions with different S_laser_ delay. We can see that the FWHM of the whole envelope curve is ~80 ps, much smaller than the value of 180 ps in Figure 5b. Meanwhile, each count distribution has an FWHM of 76 ps. It could be concluded that the timing jitter is mainly produced by the random arrival of the gating signals in the free-running mode. In the system where the delay between S_SYN_ and S_gate_ is fixed, the SPAD’s time resolution could be much enhanced.

In laser ranging systems, the number of photons returned varies for different distances and the accuracy of the distance measurement may also differ. Therefore, we evaluate the effect of different photon fluxes on the free-running SPAD’s timing jitter. The PDE is set to 2.0%. In Figure 7a, there is a slow rise in the timing jitter from 164 ps to 196 ps while the flux increases from a 0.1 photon/pulse to a 30 photon pulse. After that, the jitter begins to grow significantly, reaching 328 ps with the flux of 200 photons/pulse. We deduce that more incident photons increase the probability of producing photogenerated carriers and enlarge the avalanche time distribution. Therefore, the returned photons could be selected accordingly to achieve high-accuracy single-photon laser ranging. The operation temperature is another key parameter for avalanche photodiodes. Figure 7b displays the timing jitter as a function of the APD’s temperature. Little change in the timing jitter has taken place from −30 °C to 25 °C. The maximal jitter is 172 ps at −20 °C and 25 °C, while the minimum is 164 ps at 0 °C. In view of the TCSPC’s resolution, the timing jitters are about the same. We can conclude that the operation temperature has little effect on the free-running SPAD’s time resolution, making it more practical and easier to integrate.

## 5. Conclusions

In summary, we demonstrate a free-running single-photon detector with a GHz sinusoidally gated InGaAs/InP APD. The avalanche signal has been efficiently acquired and quickly quenched, allowing the SPAD to have a maximum count rate of up to 500 MHz. To further increase the SPAD’s maximum count rate, we could employ gating signals with higher repetition frequencies to shorten the SPAD’s dead time. Moreover, the low-pass filters with higher cutoff frequencies could be used to decrease the rise time of the avalanche signal. Additionally, the timing jitter of the SPAD has been thoroughly assessed and examined. With shorter gating pluses, the time resolution could be further improved. It can be concluded that the timing jitter is mainly produced by the random arrival of the gating signals in the free-running mode. With shorter gating pluses, the time resolution could be further improved. Changing the amplitude or frequency of the sinusoidal gating signal also offers an effective solution [15]. Such a high-performance and low-power free-running SPAD can further expand the application of single-photon detectors in the field of sensitive photodetection.

## Figures and Tables

**Figure 1 micromachines-14-00437-f001:**
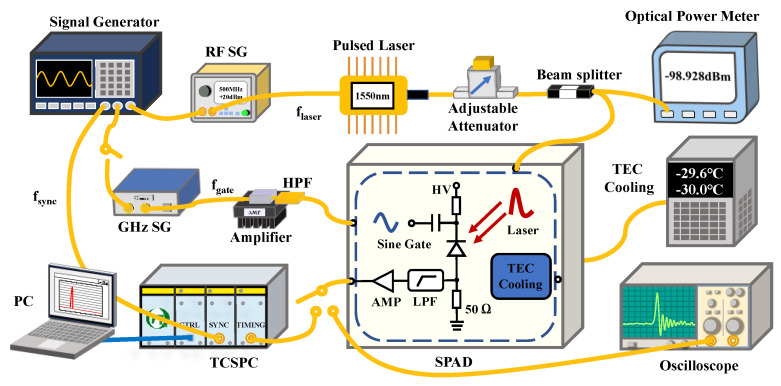
Schematic setup of the 1-GHz sinusoidally gated free-running InGaAs/InP APD. SG: signal generator; HPF: high-pass filter; LPF: low-pass filter; AMP: amplifier; TCSPC: time-correlated single-photon counter; SPAD: single-photon avalanche photodiode.

**Figure 2 micromachines-14-00437-f002:**
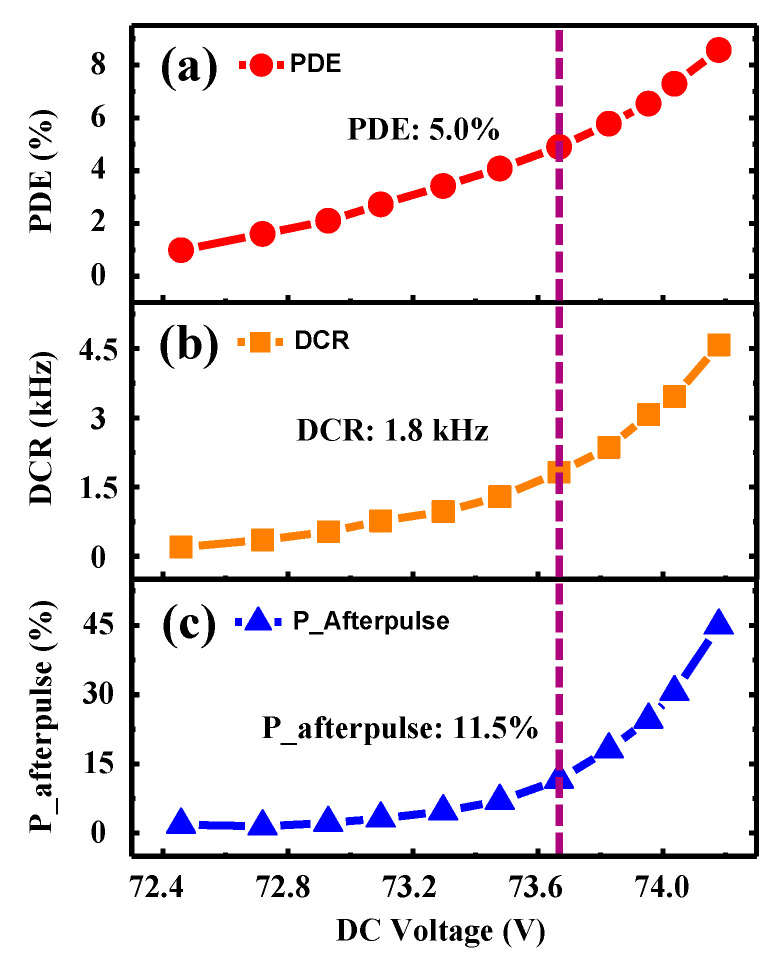
(**a**–**c**) Photon detection efficiency, dark count rate and probability of afterpulse as a function of DC bias voltage.

**Figure 3 micromachines-14-00437-f003:**
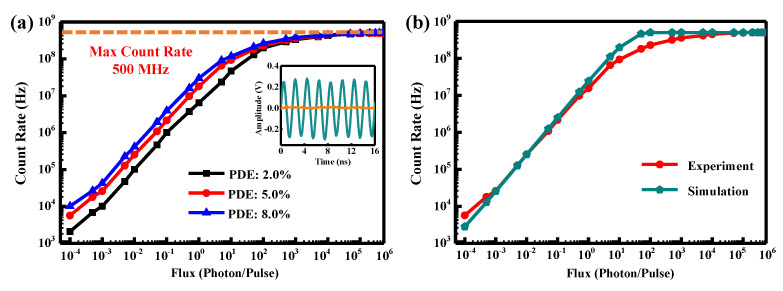
(**a**) Count rate versus flux of the incident laser with different PDEs. Inset: Waveform recorded for the SPAD under an illumination flux of 100 photons per pulse (green line) as well the residual background recorded under no illumination (orange line), respectively. (**b**) Experimental and simulated count rate as a function of the incident photon flux with the PDE of 5.0%.

**Figure 4 micromachines-14-00437-f004:**
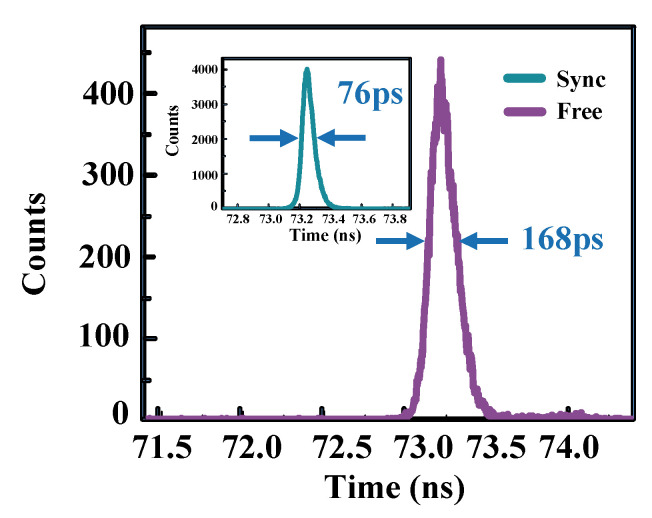
Timing jitter of the SPAD in the free-running mode. Inset: Timing jitter of the SPAD in the synchronously gating mode with the detection efficiency of 10.0%.

**Figure 5 micromachines-14-00437-f005:**
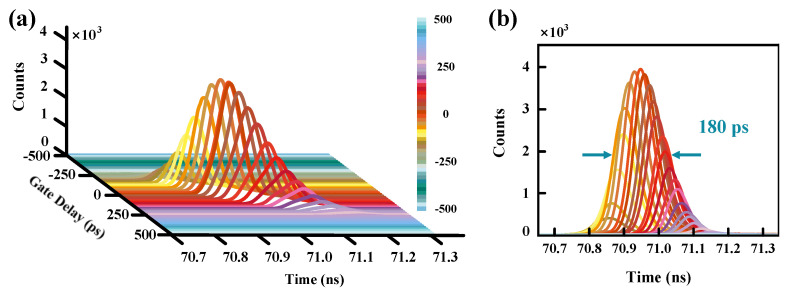
(**a**) Time distributions of the SPAD’s counts as a function of the delay between S_SYN_ and S_gate_. (**b**) Accumulation time distributions with different S_gate_ delay.

**Figure 6 micromachines-14-00437-f006:**
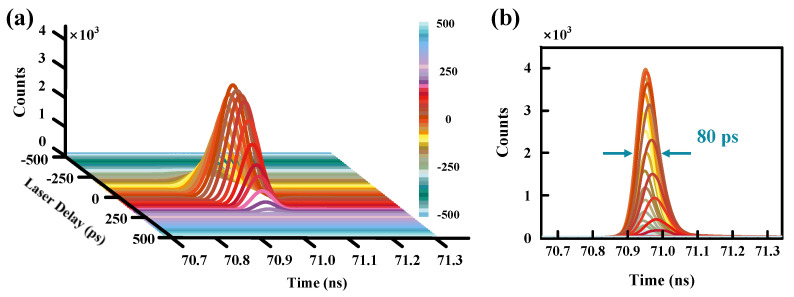
(**a**) Time distributions of the SPAD’s counts as a function of the delay between S_SYN_ and S_laser_. (**b**) Accumulation time distributions with different S_laser_ delay.

**Figure 7 micromachines-14-00437-f007:**
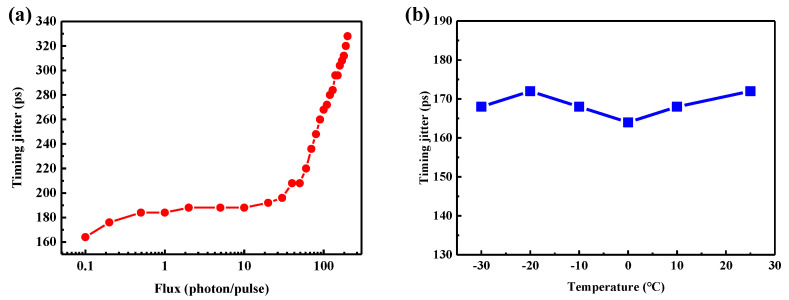
(**a**) Timing jitter as a function of incident photon flux. (**b**) Timing jitter vs. APD’s operation temperature.

## Data Availability

Not applicable.

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
