# Peer review of "Free-Running Single-Photon Detection via GHz Gated InGaAs/InP APD for High Time Resolution and Count Rate up to 500 Mcount/s"

_micromachines, 2023, doi:10.3390/mi14020437_

Round 1
Reviewer 1 Report
This work is interesting, timely and novel and should be published.
Quite a lot of text is devoted to detailed description of graphs that can be clearly seen from the graphs and so is not necessary.
The paper lacks discussion of the results and analysis of improvements that could be made in SPADs to improve jitter and count rates.
Please reference this work "N. Krstajić, S. Poland, J. Levitt, R. Walker, A. Erdogan, S. Ameer-Beg, and R. Henderson, "0.5 billion events per second time correlated single photon counting using CMOS SPAD arrays," Opt. Lett. 40, 4305-4308 (2015)."
More details of the InGaAs/InP SPAD are required. What is the structure and properties?
There needs to be some details of the 1550nm laser and its rise time, pulse width and jitter.
Line 109 "limiting the SPAD’s applications in quantum information. " Please give a value for the acceptable level of after pulse for quantum information.
Line 127 "The avalanche signal is outstanding, while the background is almost negligible, 127 implying a high SNR of the detecting scheme. " Please quantify outstanding and negligible.
Reviewer 2 Report
The authors studied the free-running single-photon detector with a GHz sinusoidally gated InGaAs/InP APD. They found that the SPAD has a maximum count rate of up to 500 MHz. Compared with the work by A. Tosi, et.al.[Alberto Tosi, Carmelo Scarcella, Gianluca Boso, and Fabio Acerbi, Gate-Free InGaAs/InP Single-Photon Detector Working at Up to 100 Mcount/s, IEEE Photonics Journal, 5(4), 6801308(2013) DOI: 10.1109/JPHOT.2013.2278526], this work has an improvement in frequency. I recommend its publication.
Author Response
We are grateful to the reviewer for his/her positive comments about our work.
Round 2
Reviewer 1 Report
Thank you for modifying the paper and adding the suggested revisions.